# RNA-Seq Transcriptome Analysis of Differentiated Human Oligodendrocytic MO3.13 Cells Shows Upregulation of Genes Involved in Myogenesis

**DOI:** 10.3390/ijms23115969

**Published:** 2022-05-25

**Authors:** Aleksandra Głowacka, Ewa Kilańczyk, Małgorzata Maksymowicz, Małgorzata Zawadzka, Wiesława Leśniak, Anna Filipek

**Affiliations:** 1Nencki Institute of Experimental Biology, Polish Academy of Sciences, 3 Pasteur Street, 02-093 Warsaw, Poland; al.glowacka@nencki.edu.pl (A.G.); m.maksymowicz@nencki.edu.pl (M.M.); m.zawadzka@nencki.edu.pl (M.Z.); w.lesniak@nencki.edu.pl (W.L.); 2Department of Medical Biology, Pomeranian Medical University in Szczecin, 1 Rybacka Street, 70-204 Szczecin, Poland; ewa.kilanczyk@pum.edu.pl

**Keywords:** MO3.13 cells, differentiation, phorbol 12-myristate 13-acetate, oligodendrocytes, RNA polymerase I (Pol I) inhibitor, p53 activity, myogenesis, K-Ras signaling

## Abstract

In this work, we examined the differentiation of oligodendrocytic MO3.13 cells and changes in their gene expression after treatment with phorbol 12-myristate 13-acetate, PMA, or with RNA polymerase I (Pol I) inhibitor, CX-5461. We found that MO3.13 cells changed their morphology when treated with both agents. Interestingly, CX-5461, but not PMA, induced noticeable changes in the integrity of the nucleoli. Then, we analyzed the p53 transcriptional activity in MO3.13 cells and found that it was increased in both cell populations, but particularly in cells treated with PMA. Interestingly, this high p53 transcriptional activity in PMA-treated cells coincided with a lower level of an unmodified (non-phosphorylated) form of this protein. Since morphological changes in MO3.13 cells after PMA and CX-5461 treatment were evident, suggesting that cells were induced to differentiate, we performed RNA-seq analysis of PMA-treated cells, to reveal the direction of alterations in gene expression. The analysis showed that the largest group of upregulated genes consisted of those involved in myogenesis and K-RAS signaling, rather than those associated with oligodendrocyte lineage progression.

## 1. Introduction

Oligodendrocytes are the source of myelin in the vertebrate central nervous system (CNS). These highly specialized cells originate from oligodendrocyte precursor cells (OPCs). In late embryonic development and shortly after birth, OPCs proliferate intensively and can migrate long distances throughout the CNS. Finally, they differentiate into myelin-producing oligodendrocytes that ensheathe the axons [1]. The myelin sheath can be destroyed as a consequence of various insults to the nervous tissue, including spinal cord injury (SCI), or as a result of inflammatory-demyelinating neurological disorders, of which multiple sclerosis (MS) is the most prevalent. Although OPCs are found to be abundant in demyelinated lesions, the process of remyelination is not efficient, most probably due to impairment in OPC differentiation.

Various cellular models, including immortalized cell lines, have been widely used to study the basis of complex biological processes, such as differentiation, aging, and tumorigenesis. Where oligodendrocyte differentiation is concerned, the human oligodendrocytic MO3.13 cell line, constructed by fusion of a rhabdomyosarcoma with a population of adult human primary oligodendrocytes [2], has most often been employed [3]. Moreover, this cell line has also served as a model to examine cellular processes implicated in a number of oligodendrocyte-linked diseases, such as multiple sclerosis (MS) [4], traumatic spinal cord injury (SCI) [5], and schizophrenia [6]. However, a recent study employing MO3.13 cells and various differentiating protocols, reported no significant changes in the expression of early or late markers of oligodendrocyte differentiation [7].

Nucleoli are the sites of rDNA transcription to rRNAs by RNA polymerase I (Pol I). Factors that interfere with this fundamental cellular process, for example inhibitors of Pol I, induce so-called ribosomal or nucleolar stress, which in turn activates signaling pathways that may lead to p53-dependent cell cycle arrest, apoptosis, differentiation, and/or senescence [8,9].

In this work, we aimed to investigate the influence of inhibition of ribosome biogenesis by a Pol I inhibitor, CX-5461, on MO3.13 cell differentiation, in comparison with the effects of treatment with PMA, a protein kinase C (PKC) activator and an established agent used to induce differentiation of MO3.13 cells [10]. Then, we searched for the mechanism responsible for CX-5461- and PMA-induced changes in MO3.13 morphology, by analyzing the nucleolar structure and p53 activity. Finally, we determined, using RNA-seq analysis, what changes in gene expression accompany the alterations in MO3.13 cell morphology.

## 2. Results

### 2.1. Influence of PMA and CX-5461 on Morphology of MO3.13 Cells

Changes in morphology may be indicative of cell differentiation, especially in the case of highly specialized cells such as oligodendrocytes. In our study, we analyzed the morphology of MO3.13 cells treated with PMA and Pol I inhibitor, CX-5461 (Figure 1). We noticed that the treated cells differed remarkably from controls. Cells treated with PMA became more spindle-like, with very long protrusions, while CX-5461-treated cells had a more compact shape, with shorter protrusions. 

Since CX-5461, a Pol I inhibitor, is supposed to induce nucleolar stress, we further compared the influence of PMA and CX-5461 on the nucleolar integrity of MO3.13 cells. For that, we applied immunocytochemical staining with the use of antibodies recognizing nucleophosmin (NPM1), a nucleolar marker, and ribosomal protein L6 (RPL6) also present in the nucleoli [11]. As it is shown in Figure 2, treatment with CX-5461, but not PMA, induced noticeable changes in the integrity of the nucleoli. In the case of PMA, staining of NPM1 and RPL6 shows their restricted nuclear localization, most probably matching that of the nucleoli. In turn, upon CX-5461 treatment, the staining of NPM1 and RPL6 was visible throughout the whole nucleus, indicating a disrupted integrity of the nucleoli. Additionally, the signal from the anti-RPL6 antibody became more intensive in the cytoplasm.

Next, we analyzed the influence of PMA and CX-5461 on p53 activity. Nucleolar proteins (e.g., RPL6) released after treatment of cells with CX-5461 are thought to bind to HDM2 and, as a result, increase p53 level by blocking its degradation [12]. However, we did not observe significant changes in total p53 level in CX-5461-treated cells in comparison with control ones (Appendix A). On the other hand, p53 level was significantly lower in PMA-treated cells when compared to both control and CX-5461-treated cells. We have subsequently performed a luciferase assay to assess p53 transcriptional activity. For this, cells were transfected with plasmid encoding firefly luciferase, under a promoter containing several binding sites for p53. Luciferase activity in CX-5461-treated cells was found to be slightly higher than in control cells (by about 40%) but, surprisingly, much lower than in PMA-treated cells (Figure 3A). 

Interestingly, this high transcriptional p53 activity in PMA-treated cells coincided with a lower reactivity with the monoclonal pAb241 antibody (Figure 3B,C), which preferentially recognizes the unmodified (a non-phosphorylated) form of p53 with low DNA binding affinity [13]. In total, treatment of MO3.13 cells with PMA or CX-5461 resulted in higher p53 transcriptional activity, which was especially evident in PMA-treated cells.

### 2.2. Expression of Genes in MO3.13 Cells after PMA Treatment

Recent work established that, although oligodendrocytic MO3.13 cells change their morphology after treatment with PMA, the expression of some important differentiation markers could only be detected at the mRNA level and, furthermore, this level did not increase significantly upon PMA treatment [7]. In the present work, we performed RT-qPCR analysis of several oligodendrocyte differentiation markers (e.g., *MBP*—*myelin basic protein*, *MOG*—*myelin-oligodendrocyte glycoprotein* and *GPR17*—*G protein-coupled receptor 17*) and found that their mRNA levels were undetectable, even after PMA treatment (not shown). The same was true for CX-5461-treated cells. Thus, to gain a broader view of the alterations in gene expression that accompany morphological changes in MO3.13 cells, we decided to perform RNA-seq transcriptome analysis of PMA-treated MO3.13 cells, the most often studied model of oligodendrocyte differentiation, and reveal the direction of the differentiation process. 

The transcriptome analysis covered 28,461 genes, of which 12,625 underwent statistically significant changes in expression after PMA treatment (6769 upregulated and 5856 downregulated genes) (Appendix A). The heat map presented in Figure 4 shows the twenty five most upregulated and twenty five most downregulated genes. 

Gene ontology analysis, performed using the GSEA software for genes with a fold change higher than 10 (1790 genes in total, 1434 upregulated and 356 downregulated), did not identify any dominant groups of differentially expressed genes (Figure 5). The largest group of upregulated genes consisted of genes involved in myogenesis (14.44%), followed by a group of genes regulated by K-Ras activation (13.90%) (Figure 5, upper panel). On the other hand, the vast majority (around 75%) of the most downregulated genes were those involved in various aspects/phases of cell division (Figure 5, lower panel). Indeed, we confirmed that PMA-treated MO3.13 cells had a lower proliferation rate (Appendix A). This is in agreement with the observed morphological changes, which suggests that the cells started to differentiate.

Regarding the large group of upregulated genes classified by GSEA as genes involved in myogenesis, Figure 6 presents a list of twenty six genes with at least 10-times increased expression. Among them there are genes encoding major contractile and regulatory muscle proteins and proteins involved in Ca^2+^ transport and storage. 

A second large group of genes with at least a 10-times higher expression consists of twenty three genes classified by GSEA as genes associated with K-Ras activation (Figure 7). 

Since the genes commonly recognized as markers of various stages of oligodendrocyte differentiation were not identified by GSEA, we read their expression status from the RNA-seq list of differentially expressed genes. Of about twenty oligodendrocyte differentiation genes that were selected based on results published by Traiffort et al. [14] and De Kleijn et al. [7], twelve were identified by RNA-seq analysis and only eight showed altered expression in PMA-treated cells. As presented in Figure 8, contrary to the myogenic genes, the changes in expression of oligodendrocyte differentiation genes were fairly moderate, with only *GPR17* and *MBP* exceeding the 10-fold change threshold. Expression of three genes, i.e., *MYRF* (*myelin regulatory factor*), *CNP*, and *VCAN* (*versican*), was downregulated following PMA-induced differentiation of MO3.13 cells.

In order to find out if MO3.13 cells treated with CX-5461 followed a similar or distinct differentiation program as those treated with PMA, we compared the changes in mRNA level of several of the most highly upregulated genes from the two top gene ontology groups i.e., myogenesis: *CASQ2* (calsequestrin 2), *TNNT3* (troponin T), *MYH* (myosin heavy chain), and K-Ras activation: *HSD11B1* (hydroxysteroid 11-beta dehydrogenase 1) by RT-qPCR. The results confirmed the enhanced expression of these genes in PMA-treated cells but, interestingly, not in CX-5461-treated ones (Figure 9). In the latter cells, the mRNA level of *CASQ2*, *TNNT3*, *MYH* (various isoforms), and *HSD11B1* was the same as in control cells. This strongly indicates that, despite visible morphological changes, the gene expression in CX-5461-treated cells was either unchanged or followed another, not yet identified, differentiation pattern.

Additionally, we performed Western blot analysis with antibodies against TNNT or HSD11B1, i.e., products of the myogenic and K-Ras activated genes, respectively. In accordance with the RT-qPCR results, this analysis showed higher levels of both proteins in PMA, but not in CX-5461-treated cells (Appendix A).

## 3. Discussion

Oligodendrocytes are highly specialized cells that originate from oligodendrocyte precursor cells (OPCs). OPCs can migrate long distances throughout the central nervous system and are able to differentiate into myelin-producing oligodendrocytes that ensheathe the axons [1]. Various cellular models, including immortalized cell lines, have been widely used to study the molecular mechanisms underlying the process of differentiation. In the case of oligodendrocyte differentiation, the human MO3.13 cell line has been commonly used in in vitro studies. Moreover, this cell line was applied to examine cellular processes implicated in a number of oligodendrocyte-linked diseases [4,5,6]. Changes in cell morphology are one of the most noticeable features of the differentiation process. It was shown by many groups that differentiation of MO3.13 cells could be induced by PMA, a PKC activator [7,10,15,16]. In addition, other agents, including T3, reactive oxygen species, D-aspartate, and serum depravation, were shown to induce MO3.13 differentiation [7,16,17,18]. In our work we analyzed the morphology of MO3.13 cells treated with PMA and Pol I inhibitor, CX-5461. Pol I inhibitor induces nucleolar stress that, in turn, activates signaling pathways, which may lead to p53-dependent cell cycle arrest, apoptosis, differentiation, and/or senescence [8,9]. Indeed, we found that MO3.13 cells treated with PMA or CX-5461 differed remarkably from untreated cells, as they developed a distinct, more ramified, morphology. As expected, treatment of MO3.13 cells with CX-5461 induced noticeable changes in the integrity of the nucleoli, as was proven by immunocytochemical staining of nucleolar proteins, NPM1 and RPL6. Some nucleolar proteins (e.g., RPL6) released after treatment of cells with CX-5461 are thought to bind to HDM2 and increase the p53 level by blocking its degradation [12]. However, our results did not show significant changes in total p53 level in CX-5461-treated cells; instead a decrease was noted in PMA-treated cells, when compared to the controls. Surprisingly, we found that p53 activity measured by the luciferase assay was highest in PMA-treated cells. This activity coincided with a lower level of unmodified p53, measured with antibodies preferentially recognizing a non-phosphorylated form of p53 with a compromised ability to bind to DNA [13]. Altogether, treatment of MO3.13 cells with PMA or CX-5461 did not induce an increase in total p53 level but resulted in higher p53 transcriptional activity; however, the nature of the p53 post-transcriptional modifications responsible for this activation requires further studies. Since p53 is intimately involved in the process of cell differentiation [19], we can presume that the slight activation of p53 by CX-5461 may not be sufficient to induce differentiation of MO3.13 cells. 

Earlier works on MO3.13 cells evaluated their appropriability as a model of oligodendrocyte differentiation, based on changes in expression of a limited number of marker genes/proteins [15,16]. On the other hand, a recent study delivered a wide range of RT-qPCR and Western blot results concerning the expression of genes associated with various stages of oligodendrocyte differentiation [7]. These results call into question the validity of using MO3.13 cells as an in vitro model of the myelination process. Thus, to unveil the direction of MO3.13 cell differentiation, we performed RNA-seq transcriptome analysis of PMA-treated cells. This analysis revealed that quite a large group of the genes upregulated in MO3.13 cells following PMA treatment consisted of genes involved in myogenesis. To this group belong genes encoding major contractile and regulatory muscle proteins, such as myosin heavy chains (MYH1/2/7), troponins (TNNT3 and TNNC2), α-actinins (ACTN2/3), and genes encoding proteins involved in Ca^2+^ transport and storage (CASQ1/2 and CaCNG1). A similar set of genes was found to be activated upon differentiation of myoblast C2C12 cells [20]. This finding is in agreement with the characteristics of MO3.13 cells. As it has already been mentioned, MO3.13 cells originate from the fusion of 6-thioguanine-resistant mutant cells of the human rhabdomyosarcoma, a type of skeletal muscle cancer, with adult human primary oligodendrocytes cultured from surgical specimens [2,3,15]. 

The second large group of upregulated genes consisted of K-Ras-dependent genes involved in various cellular signaling pathways. Among them there were genes such as *HSD11B1*, encoding an enzyme which reduces cortisone to the active hormone cortisol; *SPP1*, encoding a protein involved in the integrin and ERK1/2 signaling pathways; and *PPBP*, the product of which plays a role in DNA synthesis, mitosis, or glycolysis. Of note, mutations in K-Ras are often detected in rhabdomyosarcoma, which suggests that K-Ras signaling is also important for the proper course of myogenesis [21]. On the other hand, upregulation of K-Ras-dependent genes may also be indicative of MO3.13 cell differentiation in the direction of the oligodendrocyte lineage since ERK1/2 kinase, which is a downstream target of K-Ras, was shown to promote this process [22]. Moreover, mice deficient in R-Ras1/2, close homologs of K-Ras, showed features of hypomyelination, a higher proportion of immature to mature oligodendrocytes and lower oligodendrocyte viability, due to weaker PI3K/Akt and ERK1/2-MAPK signaling [23]. Regarding markers of various stages of oligodendrocyte differentiation [14], RNA-seq analysis identified several genes with slightly altered expression in PMA-treated MO3.13 cells. Of note, three of these, i.e., *MYRF*, *VCAN*, which are markers of the earlier stages of oligodendrocyte differentiation, and *CNP*, were downregulated. Two genes, *GPR17* and *MBP*, which like *CNP* are markers of mature oligodendrocyte, were found to be upregulated following PMA-induced differentiation of MO3.13 cells, even though their level was undetectable in RT-qPCR analysis. 

Of note, other researchers, by applying a mass spectrometry approach, have shown that the MO3.13 cell differentiation, after PKC activation by PMA, resulted in changes in the level of proteins involved in gliogenesis and some other processes, such as cytoskeletal remodeling, cell cycle, or metabolism [10]. Our analysis provides a broader view of the characteristics of MO3.13 cells and of the direction of changes in gene expression following PMA and CX-5431-treatment. It shows that the PMA-induced differentiation of MO3.13 cells triggers activation of many signaling pathways, of which the one associated with oligodendrocyte maturation does not seem to be either the most predominant or consistent. Moreover, the RT-qPCR results showed that none of the main differentiation pathways identified by RNA-seq in PMA-treated MO3.13 cells seem to be activated by CX-5461 treatment, despite evident changes in cell morphology. Altogether, in this work, we present for the first time an in-depth molecular analysis of the changes in gene expression that occur in MO3.13 cells in response to PMA treatment. Our results clearly demonstrate that this agent induces upregulation of the genes involved in myogenesis, rather than those associated with oligodendrocyte lineage progression. Thus, we claim that the MO3.13 cell line is not an appropriate model to study the oligodendrocyte differentiation and cellular processes implicated in oligodendrocyte-linked diseases.

## 4. Material and Methods

### 4.1. Cell Culture, Treatment with PMA and CX-5461, and Analysis of Cell Morphology and Proliferation

The human oligodendrocytic cell line MO3.13 was purchased from Cedarlane (Burlington, ON, Canada). Cells were cultured according to the manufacturer’s protocol in DMEM (D5796, Sigma-Aldrich, St. Louis, MO, USA) supplemented with 10% FBS (Thermo Fisher Scientific, Waltham, MA, USA), 100 μg/ml streptomycin, and 100 U/mL penicillin (both from Sigma-Aldrich, St. Louis, MO, USA) in 5% CO_2_ at 37 °C. The medium was changed every 2–3 days, and cells were passaged when confluent. To induce differentiation, MO3.13 cells were incubated in the presence of 100 nM PMA (Sigma-Aldrich, St. Louis, MO, USA) or in the presence of 200 nM Pol I inhibitor, CX-5461 (Cayman Chemical, Ann Arbor, MI, USA) in a medium containing DMEM supplemented with 1% FBS. The medium was exchanged every second day, and cell morphology was analyzed daily. Cell morphology was examined using an Axiovert 40C microscope (Carl Zeiss, Jena, Germany) equipped with an A-Plan 10×/0.25 Ph1-objective.

Proliferation of MO3.13 cells was assessed by cell counting, using an EVE automatic cell counter (NanoEnTek, Seoul, Korea). Cells were plated onto a 12-well plate at the density of 150,000 cells per well. After 24 h, cells were treated with PMA (Sigma-Aldrich, St. Louis, MO, USA) at a final concentration of 100 nM or left untreated (control). Cells were cultured for additional 72 h, then trypsinized and counted.

### 4.2. Transient Transfection of MO3.13 Cells and Luciferase Assay

To check the activity of p53 upon treatment of MO3.13 cells with PMA or CX-5461, a Dual-Luciferase Reporter Assay System (Thermo Fisher Scientific, Waltham, MA, USA) was applied. Cells were seeded on 24-well plates (50,000 cells per well) and cultured in DMEM supplemented with 1% FBS. After 24 h, cells were treated with 200 nM CX-5461 or 100 nM PMA or left untreated (control). Then, after 48 h, cells were co-transfected with the pRL-SV40 reference plasmid (Promega GmbH; Walldorf, Germany) as an internal control and plasmid containing multiple p53 binding sites cloned upstream of the firefly luciferase gene (pGl3-p53-luc). The transfection mixture was added to the medium containing DMEM and 5% FBS without antibiotics. After 5 h of incubation, the medium was changed to DMEM supplemented with 1% FBS and 200 nM CX-5461 or 100 nM PMA. Next, 24 h after transfection, cells were washed with PBS and incubated at room temperature for 15 min in Passive Lysis Buffer (Dual-Luciferase Reporter Assay System; Thermo Fisher Scientific, Waltham, MA, USA). Then, lysates were collected and luciferase activities were measured according to the manufacturer’s protocol in a Glomax 20/20 luminometer (Promega GmbH; Walldorf, Germany).

### 4.3. Immunocytochemistry

Immunofluorescent staining was performed on MO3.13 cells cultured on glass cover slips placed in a 12-well plate (2000 cells/well) in DMEM supplemented with 1% FBS. Cells, control and treated with 200 nM CX-5461 or 100 nM PMA for 3 days (4 days from seeding), were fixed with ice cold 3% paraformaldehyde (Sigma-Aldrich, St. Louis, MO, USA) for 20 min at room temperature and washed twice with PBS. Cover slips were incubated with 50 mM NH_4_Cl for 10 min at room temperature and washed again with PBS (Sigma-Aldrich, St. Louis, MO, USA) followed by treatment with ice cold Triton X-100 (0.3%) (Sigma-Aldrich, St. Louis, MO, USA) in ICCH buffer (120 mM PIPES, 50 mM HEPES, 20 mM EGTA, 8 mM MgCl_2_, pH 6.9, all reagents from Sigma-Aldrich, St. Louis, MO, USA) for 4 min. Subsequently, cells were washed twice with PBS and blocked for 1 h at 37 °C in blocking buffer containing 10% normal goat serum (Thermo Fisher Scientific, Waltham, MA, USA) and 1% BSA in PBS. Cells were stained overnight at 4 °C with mouse anti-NPM1 diluted 1:800 (Abcam, Cambridge, UK; cat. no ab10530) or rabbit anti-RPL6 diluted 1:500 (Invitrogen, Thermo Fisher Scientific, Waltham, MA, USA; cat. no PA5-30217) in 1% BSA in PBS. After three washes in PBS, cover slips were incubated for 1 h at room temperature with secondary antibodies: goat anti-rabbit IgG antibody conjugated with Alexa Fluor 488 (cat. no A-11008) and goat anti-mouse IgG conjugated with Alexa Fluor 488 (cat. no A-11029), both diluted 1:300 (both from Thermo Fisher Scientific, Waltham, MA, USA). Then, cells were washed three times with PBS and cover slips were mounted on slides with VectaShield containing DAPI (Sigma-Aldrich, St. Louis, MO, USA). Immunofluorescence staining was analyzed under a confocal microscope (LSM 800, Carl Zeiss, Jena, Germany) equipped with a 63 × oil objective at the Laboratory of Imaging Tissue Structure and Function (Nencki Institute of Experimental Biology, Warsaw, Poland).

### 4.4. Preparation of Cell Lysates, SDS-PAGE, and Western Blot 

In order to prepare protein lysates for Western blots, cells were seeded on 6-cm plates (200,000 cells per plate) and cultured in DMEM supplemented with 10% FBS. After 24 h, medium was changed for DMEM supplemented with 1% FBS, and cells were treated with 200 nM CX-5461 or 100 nM PMA or left untreated (control, Ctrl) for 72 h (for testing the level of p53) or for 120 h (for testing the level of other proteins). Then, cells were washed with PBS and incubated for 15 min on ice on a rocking platform in Passive Lysis Buffer (from Dual-Luciferase Reporter Assay System; Promega, GmbH; Walldorf, Germany), supplemented with protease inhibitor (Roche, Thermo Fisher Scientific, Waltham, MA, USA). Cells were collected in Eppendorf tubes and centrifuged for 15 min at 12,000 rpm at 4 °C (Eppendorf Centrifuge 5417R; Merck KGaA, Darmstadt, Germany). Protein concentration was measured using a Bradford assay (Bio-Rad, Hercules, CA, USA). Portions of the supernatant containing 100 μg of protein were precipitated overnight with cold acetone (−20 °C). Proteins were subjected to electrophoresis in 10% (*w*/*v*) polyacrylamide gel, performed using the method of Laemmli [24], and blotted onto nitrocellulose membrane. To analyze protein levels, the following primary antibodies were used: mouse monoclonal anti-total p53 (DO-1, Santa Cruz Biotechnology, Dallas, TX, USA) diluted 1:500, mouse monoclonal anti-unmodified p53 (pAb421, Calbiochem, Merck KGaA, Darmstadt, Germany) diluted 1:10, rabbit polyclonal anti-troponin T (#5593, Cell Signaling, Danvers, MA, USA) diluted 1:1000, and rabbit monoclonal anti-HSD11B1 (ab157223, Abcam, Cambridge, UK) diluted 1:5000.

After washing with TBS-T (50 mM Tris pH 7.5, 200 mM NaCl, 0.05% Tween 20), the blots were allowed to react with goat anti-rabbit (MP Biomedicals LLC, Irvine, Ca, USA) or goat anti-mouse (Jackson ImmunoResearch, West Grove, PA, USA) secondary antibodies conjugated to horseradish peroxidase (HRP), both diluted 1:10,000. The level of β-actin, detected by mouse monoclonal antibody conjugated with HRP (A3854, Sigma-Aldrich, St. Louis, MO, USA), diluted 1:10,000, served as an internal standard. Densitometry analysis of the detected bands was performed with ImageJ 1.42q software (NIH, Bethesda, MD, USA) [25].

### 4.5. RT-qPCR Analysis of MO3.13 Cell Differentiation Markers

Total RNA was isolated using a Universal RNA Purification kit (E3598-01, EURx, Gdańsk, Poland). One microgram of RNA was reverse transcribed using NG dART RT kit (E0801-02, EURx, Gdańsk, Poland). mRNA levels were then analyzed by RT-qPCR, by applying the SYBRGreen system with 18S rRNA as a standard. The primers used are listed in Appendix A. The results were analyzed by absolute quantification with a relative standard curve and normalized to 18S rRNA using the comparative ΔΔCt method.

### 4.6. RNA-Seq Analysis of MO3.13 Cells

Total RNA, from 3 independent experiments, was extracted from control cells and cells differentiated with 100 nM PMA for 72 h using the ExtractMe Total RNA Kit (Blirt, S.A., Gdańsk, Poland). RNA concentration was assessed by measuring absorbance using a spectrophotometer (BioSpectrometer, Eppendorf, Merck KGaA, Darmstadt, Germany). RNA sequencing was performed by the CeGaT Company (CeGaTGmbH, Tübingen, Germany). Briefly, the cDNA library was prepared using the TruSeq Stranded mRNA kit (Illumina, San Diego, CA, USA), and the sequencing was performed on a NovaSeq6000 apparatus with a Phred score of 30. After removal of adapter sequences with Skewer (Version 0.2.2.), the raw reads were aligned to hg19-cegat using STAR (Version 2.7.3). A differential expression analysis between groups, including log_2_ fold change calculation, was performed with DESeq2 (Version 1.24.0). The FDR-adjusted *p* value < 0.05 classified a significant change. The relationship between samples was visualized by hierarchical clustering and principal component analysis (PCA).

### 4.7. Statistical Analysis

Each experiment was performed in at least two biological repetitions. Data was analyzed in Prism 6 (GraphPad Software). To check the significance of differences between control (Ctrl) and PMA- or CX-5461-treated cells, observed in Western blot and luciferase assays, an independent sample *t* test was used. The significance of mean comparison is annotated as follows: * *p* ≤ 0.05, ** *p* ≤ 0.01, and *** *p* ≤ 0.001.

## Figures and Tables

**Figure 1 ijms-23-05969-f001:**
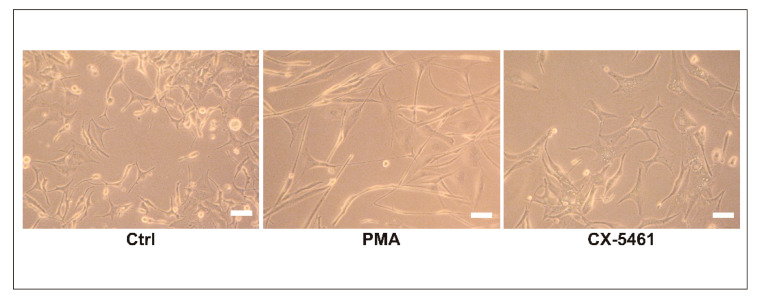
Morphology of MO3.13 cells. Representative images of control cells (Ctrl) and those treated with PMA or CX-5461 taken on the 6th day of treatment. Scale bar is 20 µm.

**Figure 2 ijms-23-05969-f002:**
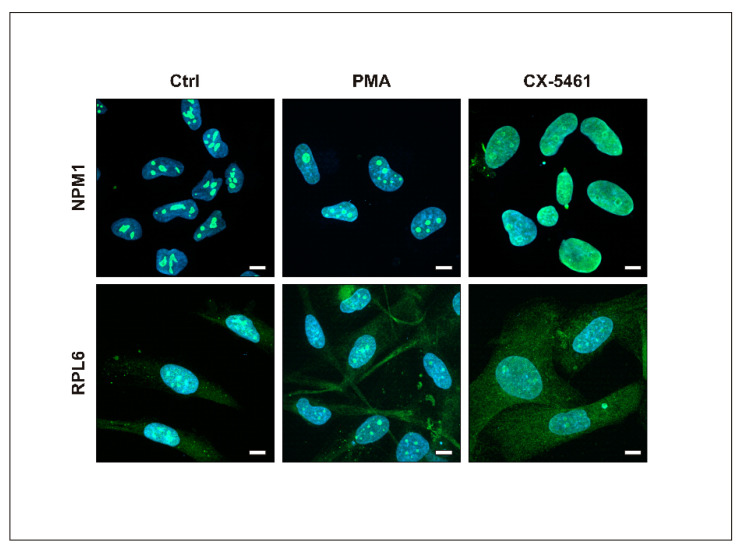
Assessment of nucleolar integrity of MO3.13 cells. Control cells (Ctrl) and cells treated with PMA or CX-5461. Representative images after staining with specific antibodies against NPM1 (green, upper panel) and RPL6 (green, lower panel) and DAPI (cyan, both panels). Scale bar is 10 µm.

**Figure 3 ijms-23-05969-f003:**
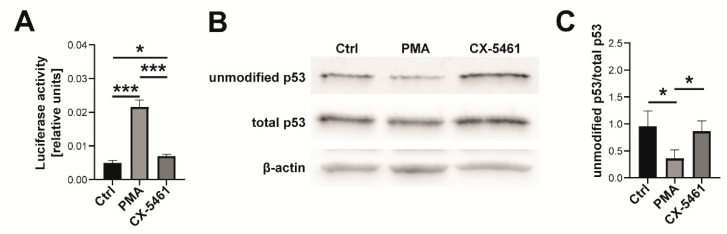
Influence of PMA and CX-5461 on p53 activity. (**A**) Luciferase assay performed using lysates from control (Ctrl) undifferentiated MO3.13 cells and cells treated with PMA or CX-5461. Results are presented as a ratio between Firefly and Renilla luciferase activity. (**B**) A representative Western blot showing protein level of unmodified p53, total p53, and actin (loading control). (**C**) Densitometric analysis of the level of unmodified p53. Results are presented as the ratio of unmodified to total p53, both normalized to actin. (**A**,**C**) Statistical analysis of results (*n* = 3) was performed with the use of one sample *t* test. Results are presented as means ± standard error. The level of statistical significance is indicated using * *p* ≤ 0.05, *** *p* ≤ 0.001.

**Figure 4 ijms-23-05969-f004:**
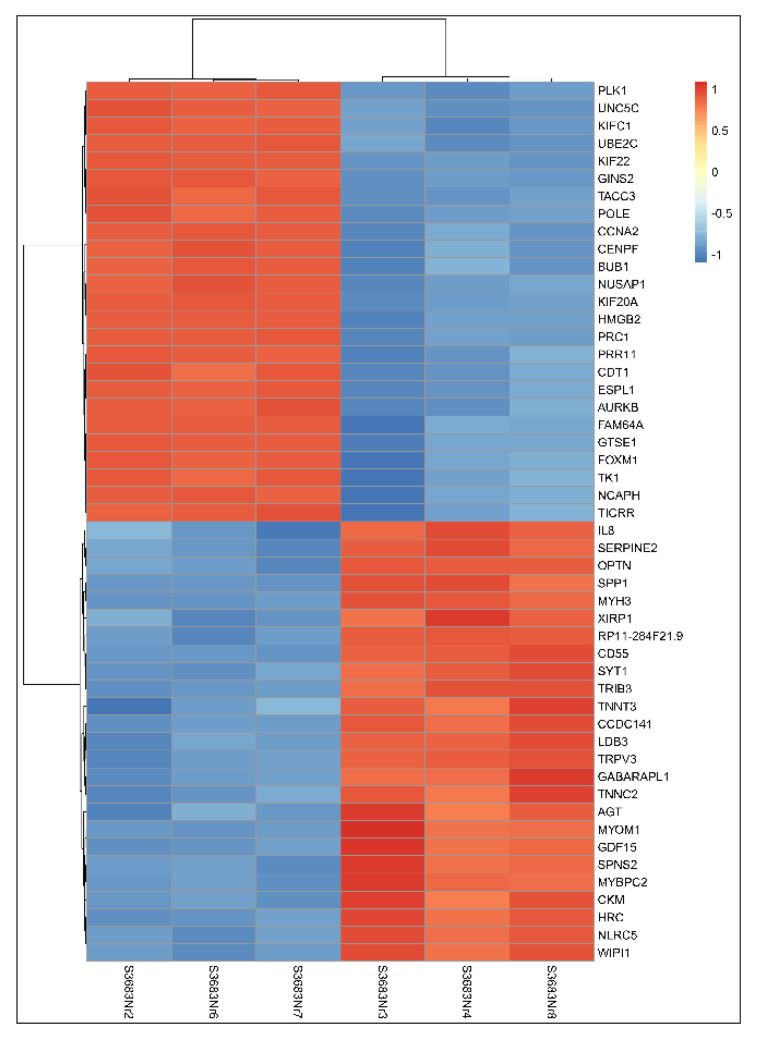
Differential gene expression analysis of RNA-seq results. Heat map showing the 50 most differentially expressed genes (25 upregulated, 25 downregulated) in three control (left side) and three PMA-treated (right side) MO3.13 cells. The columns represent individual samples. To create the heat map, the Z-score of normalized counts was used.

**Figure 5 ijms-23-05969-f005:**
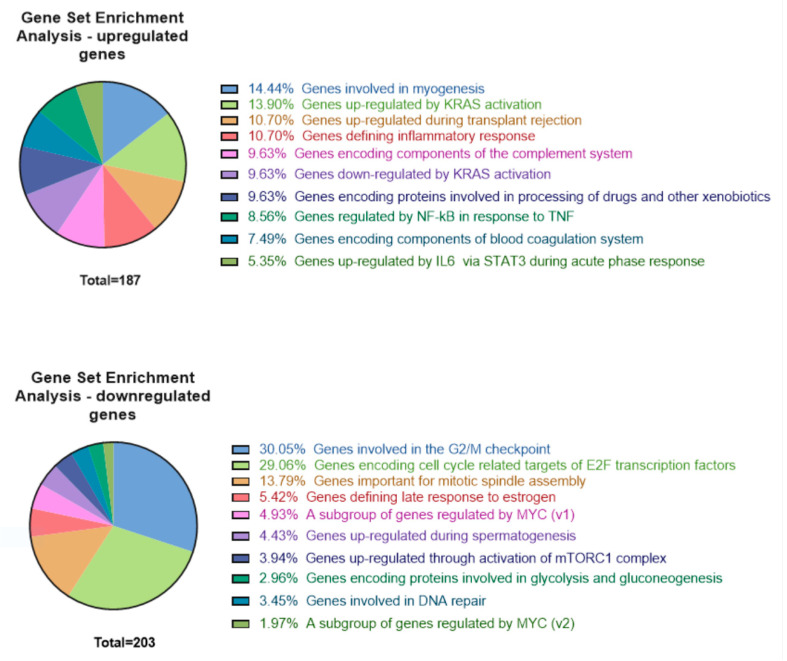
Distribution of hallmark genes that represent well-defined biological states or processes. Diagrams present the distribution of 1434 significantly upregulated and 356 downregulated genes (fold change ≥ |10|). Analysis was performed using the GSEA (Gene Set Enrichment Analysis) online tool. Numbers below the pie charts show the number of genes with the assigned function hit.

**Figure 6 ijms-23-05969-f006:**
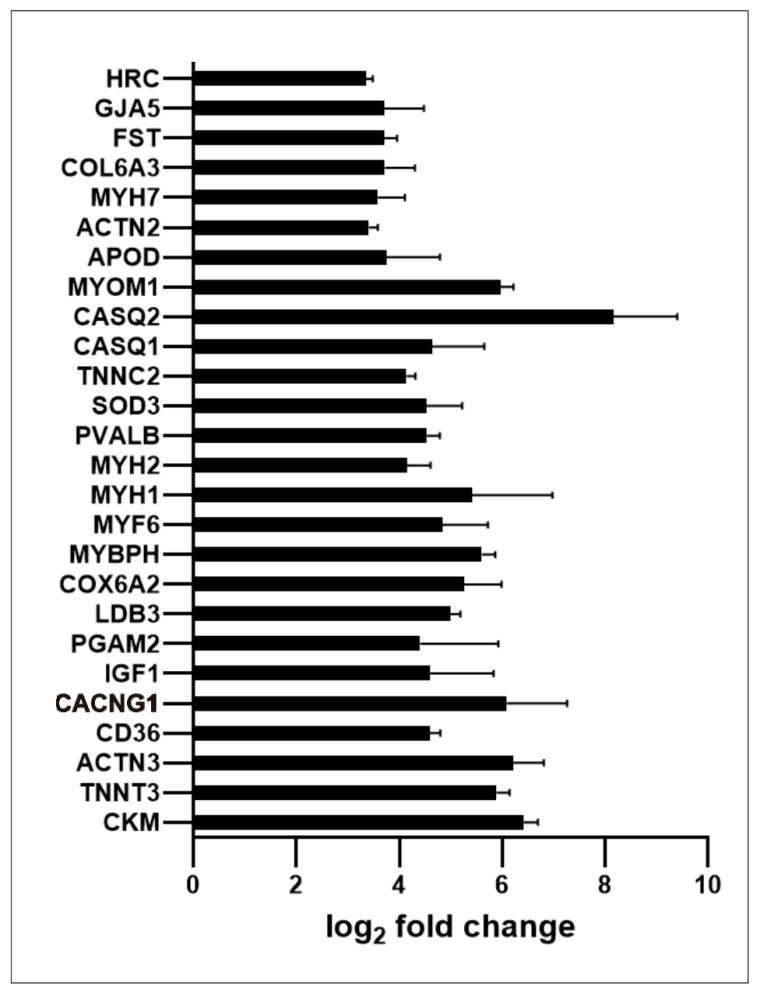
Expression of genes involved in myogenesis in PMA-treated MO3.13 cells; 26 genes with fold change ≥ |10| are shown.

**Figure 7 ijms-23-05969-f007:**
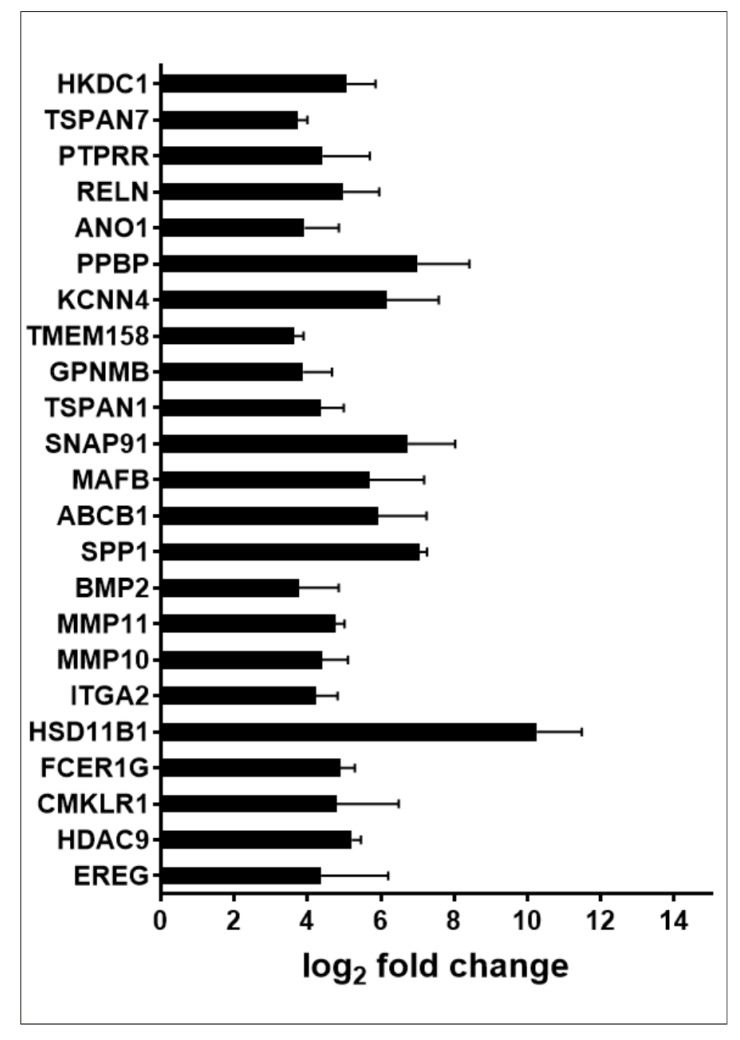
Expression of genes upregulated by K-Ras activation in PMA-treated MO3.13 cells; 23 genes with fold change ≥ |10| are shown.

**Figure 8 ijms-23-05969-f008:**
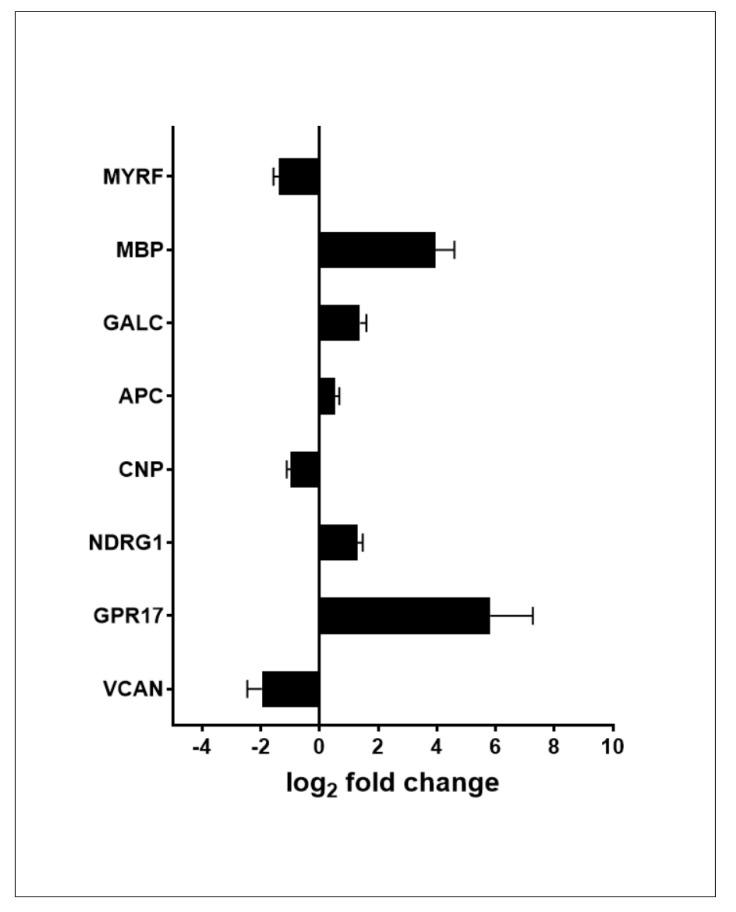
Expression of genes encoding oligodendrocyte differentiation markers in PMA-treated MO3.13 cells. Only genes with a statistically significant change in expression are shown.

**Figure 9 ijms-23-05969-f009:**
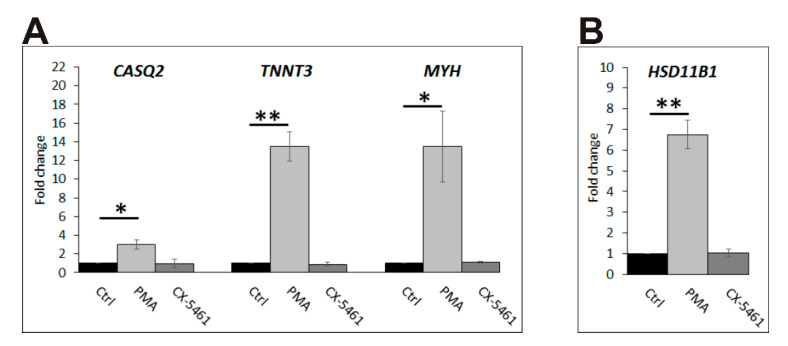
Relative mRNA expression of selected genes in MO3.13 cells. (**A**) Expression of *CASQ2*, *TNNT3*, and *MYH* (*various isoforms*). (**B**) Expression of *HSD11B1*. Control cells (Ctrl—black bars); cells treated with PMA or CX-5461, grey and dark grey bars, respectively. (*n* = 3) ** p* ≤ 0.05; *** p* ≤ 0.01.

## Data Availability

All data generated or analysed during this study are included in this published article and Appendix A. Further inquiries can be directed to the corresponding author.

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
