# Peer review of "RNA-Seq Transcriptome Analysis of Differentiated Human Oligodendrocytic MO3.13 Cells Shows Upregulation of Genes Involved in Myogenesis"

_ijms, 2022, doi:10.3390/ijms23115969_

Round 1

Reviewer 1 Report

While the revised manuscript has addressed some of my previous comments, I agree with other reviewers that the scope of the current study is limited and lacks novelty. The conclusion of the current study remains as “MO3.13 cell line is not a proper one to study oligodendrocyte differentiation”, which has been demonstrated by multiple groups and publications.

A few of my other comments are below.

It still remains unclear why RNA-seq is only performed on PMA-treated cells, not on CX-5461 treated cells.

Furthermore, given the lack of change or characterization on the properties of “differentiation”, the authors should restrict the use of PMA or CX-5461 as “differentiating agent”.

The following statement is very confusing. “However, the expression of some important differentiation markers (e.g. MBP, myelin basic protein) could only be detected at the mRNA level and….. In the present work we have performed RT-qPCR analysis of several oligodendrocyte differentiation markers … and obtained similar results i.e., the mRNA level was undetectable (not shown).”  What does it mean that transcript can only be detected at the mRNA level? By qRT-PCR?

Author Response

Reviewer 1

Comment:

While the revised manuscript has addressed some of my previous comments, I agree with other reviewers that the scope of the current study is limited and lacks novelty. The conclusion of the current study remains as “MO3.13 cell line is not a proper one to study oligodendrocyte differentiation”, which has been demonstrated by multiple groups and publications.

Answer:

We cannot fully agree with the opinion that the unsuitability of MO3.13 cells to study oligodendrocyte differentiation “has been demonstrated by multiple groups and publications”. On the contrary, there is only one comprehensive report concerning this problem (De Kleijn et al., Cells 2019, 8, 1096) and the MO3.13 cells are still used in studies on e.g.  remyelination (Velasco-Estevez  et al. Int. J. Mol. Sci. 2021, 22, 4342). This is probably more due to their origin than molecular properties since, in most cases, only single or a few oligodendrocyte marker genes are studied to prove their utility. In our manuscript we show for the first time an in-depth molecular analysis of changes in gene expression that occur in MO3.13 cells in response to PMA treatment. Our results clearly show that the most up-regulated genes in PMA-treated MO3.13 cells are genes involved in myogenesis and not in oligodendrocyte differentiation; based on this, these cells should not be used as a model of the myelination process. To underline the importance of our findings we added to the revised version of our manuscript an appropriate sentence at the end of Discussion.

Comment:

It still remains unclear why RNA-seq is only performed on PMA-treated cells, not on CX-5461 treated cells.

Answer:

Based on the RT-qPCR analysis, we found that the CX-5461 treatment was not more efficient than PMA in inducing MO3.13 expression of oligodendrocyte genes. Thus, and also because of the costs of RNA-seq analysis, we thought it more worthwhile and important to fully evaluate the PMA-treated cells since PMA is an established and a commonly used agent to induce morphological changes in MO3.13 cells that are believed to reflect their differentiation.

Comment:

Furthermore, given the lack of change or characterization on the properties of “differentiation”, the authors should restrict the use of PMA or CX-5461 as “differentiating agent”.

Answer:

This has been corrected throughout the text.

Comment:

The following statement is very confusing. “However, the expression of some important differentiation markers (e.g. MBP, myelin basic protein) could only be detected at the mRNA level and….. In the present work we have performed RT-qPCR analysis of several oligodendrocyte differentiation markers … and obtained similar results i.e., the mRNA level was undetectable (not shown).”  What does it mean that transcript can only be detected at the mRNA level? By qRT-PCR?

Answer:

This statement has been corrected/modified in the rvised version of the manuscript.

Thank you for your comments that improved the quality of our manuscript.

Reviewer 2 Report

The reviewed manuscript contains a very well performed and clearly described analysis of the applicability of the artificially immortalized cell lines as “supplements” of natural cell populations in the research. The authors performed a wide range of experiments employing diversified but complementary methods including very modern molecular techniques. The presented results arise no methodological doubts, are clearly described and discussed. And moist importantly, these findings support strongly the study conclusion that MO3.13 cell line is not a proper model to study oligodendrocyte differentiation. Thus the manuscript represents a very important contribution to the current knowledge in the field of molecular and cellular biology, neuroscience and medicine.

Author Response

Reviewer 2

Comment:

The reviewed manuscript contains a very well performed and clearly described analysis of the applicability of the artificially immortalized cell lines as “supplements” of natural cell populations in the research. The authors performed a wide range of experiments employing diversified but complementary methods including very modern molecular techniques. The presented results arise no methodological doubts, are clearly described and discussed. And most importantly, these findings support strongly the study conclusion that MO3.13 cell line is not a proper model to study oligodendrocyte differentiation. Thus the manuscript represents a very important contribution to the current knowledge in the field of molecular and cellular biology, neuroscience and medicine.

Answer:

Thank you for appreciating the importance of the results described in our manuscript.

Reviewer 3 Report

Comments

1. Lines 249-251. Lines 249-251. Why didn't the authors study cells for proliferation?

2. One wonders if culturing PMA-treated MO3.13 cells in myogenic differentiation medium leads to the formation of myotubes, or if the simultaneous upregulation of genes regulated by K-Ras activation reverts MO3.13 cells to rhabdomyosarcoma cells.

Author Response

Comment:

Lines 249-251. Lines 249-251. Why didn't the authors study cells for proliferation?

Answer:

Thank you for this comment. We have studied MO3.13 cell proliferation and found that the cells slowed down proliferation in the presence of PMA. The results are shown in Supplementary file (Figure S2).

Comment:

One wonders if culturing PMA-treated MO3.13 cells in myogenic differentiation medium leads to the formation of myotubes, or if the simultaneous upregulation of genes regulated by K-Ras activation reverts MO3.13 cells to rhabdomyosarcoma cells.

Answer:

MO3.13 cells were cultured for 6 days in the medium containing 1% FBS which corresponds to the serum concentration in myogenic differentiation medium (Moran JL et al., Physiol Genomics, 2002). As it is shown in Figure 1 of our manuscript, addition of PMA altered cell morphology. The cells became more elongated but no evident myotube formation could be observed.

Activation of K-Ras dependent genes, and not of oligodendrocyte differentiation genes, clearly indicates that the rhabomyosarcoma genetic component of MO3.13 cells is stimulated upon PMA treatment. However, it is hard to establish to what extent the MO3.13 cells revert to the rhabdomyosarcoma phenotype since this would require a detailed analysis of both cell lines.

Reviewer 4 Report

The present paper is well written and designed. The experiments were performed in a rigorous manner. I suggest to the authors to validate some genes using a different type of methodology, for e.g. WB analyses or others.

Author Response

Comment

The present paper is well written and designed. The experiments were performed in a rigorous manner. I suggest to the authors to validate some genes using a different type of methodology, for e.g. WB analyses or others.

Answer;

Thank you for your good opinion of our manuscript. As to the comment, we would like to stress that in Figure 9 of the manuscript we showed validation of the expression of selected genes (CASQ2, TNNT3, MYH, HSD11B1) from RNA-seq analysis by RT-qPCR. Additionally, we have included Western blot analyses for proteins encoded by TNNT and HSD11B1 genes in the Supplementary file (Figure S3, in the former version it was Figure S2).

Round 2

Reviewer 1 Report

The revised manuscript has addressed some of my previous concerns. However, despite the use of a more sophisticated methodology, the study has provided incremental knowledge from the current understanding in the field. The impact and significance of the study remains limited.

Author Response

Comment:

The revised manuscript has addressed some of my previous concerns. However, despite the use of a more sophisticated methodology, the study has provided incremental knowledge from the current understanding in the field. The impact and significance of the study remains limited.

Answer:

Thank you for acknowledging improvements made in the revised version of our manuscript. Our comprehensive studies provide the most profound transcriptomic analysis of MO3.13 cells and thus clearly indicate that another, a more suitable cellular model to study oligodendrocyte differentiation should be considered.

Reviewer 3 Report

The authors have adequately addressed my concerns.

Author Response

Comment:

The authors have adequately addressed my concerns.

Answer:

Thank you very much for accepting our answers.